# Effects of Coexisting Anions on the Formation of Hematite Nanoparticles in a Hydrothermal Process with Urea Hydrolysis and the Congo Red Dye Adsorption Properties

Takahiro Onizuka [1], Mikihisa Fukuda [2] and Tomohiro Iwasaki [2,*]

[1]  Department of Chemical Engineering, Osaka Prefecture University, Osaka 599-8531, Japan
[2]  Department of Chemical Engineering, Osaka Metropolitan University, Osaka 599-8531, Japan
*   Correspondence: tomohiro.iwasaki@omu.ac.jp

**Abstract:** Crystalline hematite nanoparticles as adsorbents for anionic Congo red dye were prepared by a hydrothermal process using urea hydrolysis. To examine the effects of coexisting anions in a solution on the formation of hematite nanoparticles, different iron(III) salts, including iron chloride hexahydrate, iron nitrate nonahydrate, iron sulfate *n*-hydrate, ammonium iron sulfate dodecahydrate, and basic ferric acetate, were employed as iron-ion sources. After the hydrothermal treatment of the solution, consisting of an iron salt and urea at 423 K for 20 h, a single phase of hematite was formed from the iron-nitrate solution. The results suggested that the hydrothermal formation of hematite depended on the stability of iron complexes formed in the starting solution. The average crystallite size and median diameter of hematite nanoparticles also depended on the coexisting anions, suggesting that the appropriate selection of the coexisting anions in the starting solution can allow for control of the crystallite size and particle diameter of hematite nanoparticles. The Congo red adsorption kinetics and isotherms of the hematite nanoparticles were described by the Elovich model and Langmuir model, respectively. The adsorption thermodynamics parameters were estimated, which suggested an exothermic and spontaneous process. The results demonstrated good adsorption properties for Congo red adsorption.

**Keywords:** hematite nanoparticle; urea hydrolysis; hydrothermal synthesis; coexisting anion; anionic dye adsorption

## 1. Introduction

Elemental iron is an abundant natural resource contained in the earth's crust and is conventionally and widely used in many industries as a basic material in metallic iron, steel, alloys, oxides, and hydroxides because of its low cost, high usability, and nontoxicity [1]. Among iron-containing materials, iron oxides, such as FeO, $\alpha$-, $\beta$-, $\gamma$-, and $\varepsilon$-Fe$_2$O$_3$, and Fe$_3$O$_4$, have simple crystal structures and excellent properties, resulting in many practical uses in various industrial products, such as colorants, catalysts, and magnetic materials [2,3]. In particular, hematite ($\alpha$-Fe$_2$O$_3$) fine powder has been traditionally used as a pigment; in recent years, hematite nanoparticles have been actively studied for advanced applications, e.g., as anode materials in lithium-ion batteries [4–8], adsorbents for removal of heavy metals and dyes [8–12], sensors [13–17], and photocatalysts [18–21].

Specific crystallinity and particle size are required for each application of $\alpha$-Fe$_2$O$_3$ nanoparticles and are achievable through various methods [22,23]. Among them, synthesis processes using liquid phase reactions via formation of ferric hydroxides as precursors and/or intermediates, e.g., sol–gel and hydrothermal processes with/without postcalcination, have been industrially employed due to their simplicity of operation [10,13,17,24,25]. In addition, these methods can be used to control the crystallinity and particle size of products by adjusting reaction conditions, such as temperature, concentration, and additives.

Accordingly, the liquid-phase processes are advantageous in the industrial production of $\alpha$-Fe$_2$O$_3$ nanoparticles.

As an effective method to provide homogeneous metal-oxide nanoparticles, hydrothermal treatments of hydroxides as precursors/intermediates formed by urea hydrolysis in an aqueous phase have attracted much attention; the hydroxide-ion concentration increases uniformly at temperatures higher than approximately 90 °C [26–28], resulting in the formation of homogeneous hydroxides. Some researchers have employed this method for the synthesis of $\alpha$-Fe$_2$O$_3$ nanoparticles [29–31]. In the processes, before urea hydrolysis, ferric ions can be converted to iron complexes with coexisting anions as ligands in a solution; this may affect the formation of ferric hydroxide [32,33]. This paper first reports a systematic study on the effects of coexisting anions on the formation of $\alpha$-Fe$_2$O$_3$ nanoparticles in a hydrothermal process with urea hydrolysis. To vary the coexisting anions, iron(III) salts with different anions were used as the ferric-ion source, and the hydrothermal formation of homogeneous $\alpha$-Fe$_2$O$_3$ nanoparticles was investigated.

As an application of $\alpha$-Fe$_2$O$_3$ nanoparticles, removal of harmful pollutants from wastewaters by adsorption has been conducted using $\alpha$-Fe$_2$O$_3$ nanoparticles as adsorbents [34,35]. For example, large amounts of wastewaters containing toxic organic dyes, which can have a serious impact on aquatic flora and fauna as well as human beings, have been generated in various dye industries such as dye manufacturing, textile dyeing, and printing, and discharged after proper treatments. Adsorption onto $\alpha$-Fe$_2$O$_3$ nanoparticles has often been employed as an effective treatment method to remove the pollutants from aqueous solutions [10,36–38]. Anionic Congo red dye, which is widely used in not only the textile and optoelectronics industries but also amyloidosis studies [39], can be toxic because of its carcinogenic nature due to its degradation products such as benzidine [39,40]. Although Congo-red-containing wastewaters have been purified by various methods (e.g., degradation [41–44], flocculation [45,46]), adsorption has been attracting much attention because of its simplicity and efficiency [47,48]. In recent years, $\alpha$-Fe$_2$O$_3$-based adsorbents with controlled structures have been developed for effective removal of Congo red [49,50]; however, few studies on the adsorption properties of pure $\alpha$-Fe$_2$O$_3$ nanoparticles without additional functionalization have been reported [51], although they can be simply prepared under environmentally friendly conditions and are expected to have good adsorption performance. In particular, the thermodynamic analysis of the adsorption using an appropriate adsorption equilibrium constant [52] is insufficient. Therefore, we studied the adsorption removal of Congo red dye from aqueous solutions using simply synthesized $\alpha$-Fe$_2$O$_3$ nanoparticles, and the adsorption properties were carefully investigated.

## 2. Materials and Methods

### 2.1. Hydrothermal Synthesis and Characterization of $\alpha$-Fe$_2$O$_3$ Nanoparticles

All reagents were used without further purification. As iron(III) ion sources, iron chloride hexahydrate (FeCl$_3$·6H$_2$O), iron nitrate nonahydrate (Fe(NO$_3$)$_3$·9H$_2$O), iron sulfate *n*-hydrate (Fe$_2$(SO$_4$)$_3$·$n$H$_2$O, $n$ = 7.3), and ammonium iron sulfate dodecahydrate (FeNH$_4$(SO$_4$)$_2$·12H$_2$O), purchased from FUJIFILM Wako Pure Chemicals (Osaka, Japan), and basic ferric acetate (Fe(OH)(CH$_3$COO)$_2$; Kishida Chemical, Osaka, Japan) were used. In a typical synthesis, 100 mL of a solution containing 5 mmol of iron(III) ions were prepared by dissolving an iron salt in deionized water, and a predetermined amount of urea ((NH$_2$)$_2$CO; FUJIFILM Wako Pure Chemicals, Osaka, Japan) was added to the solution. The resulting solution was hydrothermally treated at 423 K for 20 h under autogenous pressure (approximately 0.6 MPa) in a Teflon-lined stainless steel autoclave (custom-made high-pressure vessel) with a capacity of 500 cm$^3$. The solution occupied approximately 20% of the vessel capacity. During the hydrothermal treatment, $\alpha$-Fe$_2$O$_3$ nanoparticles were formed according to the following possible reactions.

$$(NH_2)_2CO + 3H_2O \rightarrow 2NH_4^+ + 2OH^- + CO_2 \tag{1}$$

$$Fe^{3+} + 3OH^- \rightarrow Fe(OH)_3 \tag{2}$$

$$Fe(OH)_3 \rightarrow \alpha\text{-FeOOH} + H_2O \tag{3}$$

$$2\alpha\text{-FeOOH} \rightarrow \alpha\text{-Fe}_2O_3 + H_2O \tag{4}$$

Thus, the overall reaction for the formation of $\alpha\text{-Fe}_2O_3$ was expressed from Equation (1) to Equation (4) as follows.

$$Fe^{3+} + (3/2)(NH_2)_2CO + (9/2)H_2O \rightarrow (1/2)\alpha\text{-Fe}_2O_3 + 3NH_4^+ + (3/2)CO_2 + (3/2)H_2O \tag{5}$$

The amount of urea in the starting solution was determined to be 37.5 mmol, which was 5 times the stoichiometric ratio stated in Equation (5). The obtained precipitate was washed with deionized water several times and dried at 353 K overnight in air. To examine the effects of coexisting anions on $\alpha\text{-Fe}_2O_3$ formation, the hydrothermal synthesis was conducted in the presence of either sodium nitrate ($NaNO_3$) or sodium sulfate ($Na_2SO_4$) purchased from FUJIFILM Wako Pure Chemicals (Osaka, Japan). Furthermore, to confirm the effects of urea hydrolysis on $\alpha\text{-Fe}_2O_3$ formation, an ammonia solution (FUJIFILM Wako Pure Chemicals, Osaka, Japan) was used instead of urea at the same ammonium ion concentration. In the experiment, 5 mmol of $Fe(NO_3)_3$, as the iron source, was dissolved in 25 mL of deionized water. Then, 75 mL of 1 mol/L ammonia solution were added dropwise to the $Fe(NO_3)_3$ solution under vigorous stirring at room temperature. The resulting suspension was hydrothermally treated for 20 h at 423 K.

The phase evolution of the samples was measured with an X-ray diffractometer (XRD-6100, Shimadzu, Kyoto, Japan) using Cu–K$\alpha$ (30 kV, 30 mA) at 1°/min. The morphology and particle size distribution were determined for samples with a single $\alpha\text{-Fe}_2O_3$ phase by scanning electron microscopy (JSM-6700F, JEOL, Tokyo, Japan) and dynamic light scattering (Zetasizer Nano ZS, Malvern Panalytical, Malvern, UK), respectively. The zeta potential was also measured for the dilute suspensions with different pH values using the same analyzer (Zetasizer Nano ZS).

### 2.2. Batch Adsorption Studies for Congo Red

Congo red (Nacalai Tesque, Kyoto, Japan, Figure 1) was used as adsorbate. The $\alpha$-$Fe_2O_3$ nanoparticles synthesized using $Fe(NO_3)_3 \cdot 9H_2O$ as the iron source were employed as the adsorbent. Typically, 10 mg of the adsorbent was added to 10 mL of an aqueous solution of Congo red with an initial concentration of 100 mg/L (pH = 7.3 without adjustment). The suspension was sonicated for 10 min and then kept at 293 K under static conditions for a predetermined amount of time (described later). After that, the adsorbent was centrifuged and the absorbance of the supernatant was measured with a spectrophotometer (U-2900, Hitachi High-Technologies, Tokyo, Japan) at a wavelength of 498 nm. The concentration $C_t$ (mg/L) at contact time $t$ (h) was determined based on the absorbance. The removal efficiency $R$ (%) and the adsorbed amount $q_t$ (mg/g) were calculated by Equations (6) and (7), respectively.

$$R = \frac{C_0 - C_t}{C_0} \times 100 \tag{6}$$

$$q_t = \frac{(C_0 - C_t)V}{m} \tag{7}$$

where $C_0$ (mg/L) is the initial concentration, $V$ (L) is the volume of solution, and $m$ (mg) is the mass of the adsorbent. The equilibrium adsorption capacity $q_e$ (mg/g) was calculated from the equilibrium concentration $C_e$ (mg/L) obtained after contact for more than 12 h using Equation (7).

To examine the effect of adsorption conditions on the removal efficiency, the dosage of adsorbent and the initial pH of aqueous solution were varied between 5 mg and 25 mg and between 3.3 and 10.7, respectively. The pH was adjusted by using 0.1 mol/L HCl or 0.1 mol/L NaOH. In the adsorption kinetics analysis, the contact time was varied from 1 h to 168 h under fixed conditions of 10 mg of adsorbent and 10 mL of 100 mg/L solution. For the adsorption isotherm analysis, the initial concentration was varied between 10 mg/L

and 100 mg/L. Furthermore, to analyze thermodynamically the adsorption process, the temperature was changed from 293 K to 313 K.

**Figure 1.** Chemical structure of Congo red dye.

## 3. Results and Discussion

### 3.1. Hydrothermal Synthesis of α-Fe₂O₃ Nanoparticles with Urea Hydrolysis

3.1.1. Effects of the Reactants on the $\alpha$-Fe$_2$O$_3$ Formation

The X-ray diffraction (XRD) patterns of the samples prepared with the different iron salts are shown in Figure 2. The phase evolution in the hydrothermal process strongly depended on the iron salts employed, i.e., the anions in solution; however, the pH of the solutions after hydrothermal treatment was 9.2 ± 0.1 regardless of the iron salts that were present. When Fe(NO$_3$)$_3$ was used, single-phase crystalline $\alpha$-Fe$_2$O$_3$ was obtained. However, samples prepared with the other iron salts contained not only $\alpha$-Fe$_2$O$_3$ but also the intermediate $\alpha$-FeOOH. The results suggested that NO$_3^-$ ions promoted $\alpha$-Fe$_2$O$_3$ formation. The hydrothermal synthesis using Fe(NO$_3$)$_3$ confirmed the complete $\alpha$-Fe$_2$O$_3$ formation reaction in a relatively short period of time (approximately 2 h), as shown in Figure 3. In this case, instead of $\alpha$-FeOOH, 6-line ferrihydrite (Fe$_5$HO$_8\cdot$4H$_2$O) intermediate [53] was detected at the early stages, which may have contributed toward the rapid formation of $\alpha$-Fe$_2$O$_3$. The results revealed that Fe(NO$_3$)$_3$ may be a suitable iron source for the hydrothermal process with urea hydrolysis.

SEM images and particle size distributions of the samples prepared with urea and the ammonia solution are shown in Figure 4. While the XRD analysis confirmed that a uniform $\alpha$-Fe$_2$O$_3$ phase was also obtained when using the ammonia solution, the particle size of $\alpha$-Fe$_2$O$_3$ nanoparticles prepared with urea was relatively uniform compared with the case using the ammonia solution. The results demonstrated that the hydrothermal process using urea hydrolysis was effective for the preparation of uniformly sized $\alpha$-Fe$_2$O$_3$ nanoparticles.

3.1.2. Change in the Crystallite Size and Particle Diameter with Coexisting Anion

To examine the reaction-promoting effect of NO$_3^-$ ions, 75 mmol of NaNO$_3$, i.e., five-times the amount of NO$_3^-$ ions contained in the starting Fe(NO$_3$)$_3$ solution, were added to the starting solutions prepared with the iron salts except for Fe(NO$_3$)$_3$; hydrothermal treatment was then performed under the same conditions. The XRD patterns of the samples prepared in the presence of NO$_3^-$ ions are shown in Figure 5. When the NaNO$_3$-added FeCl$_3$ and Fe(OH)(CH$_3$COO)$_2$ solutions were used, a single $\alpha$-Fe$_2$O$_3$ phase was obtained; in particular, high-crystalline $\alpha$-Fe$_2$O$_3$ was formed from the NaNO$_3$-added FeCl$_3$ solution. In contrast, the addition of NaNO$_3$ to the Fe$_2$(SO$_4$)$_3$ and FeNH$_4$(SO$_4$)$_2$ solutions negligibly improved the hydrothermal formation of $\alpha$-Fe$_2$O$_3$.

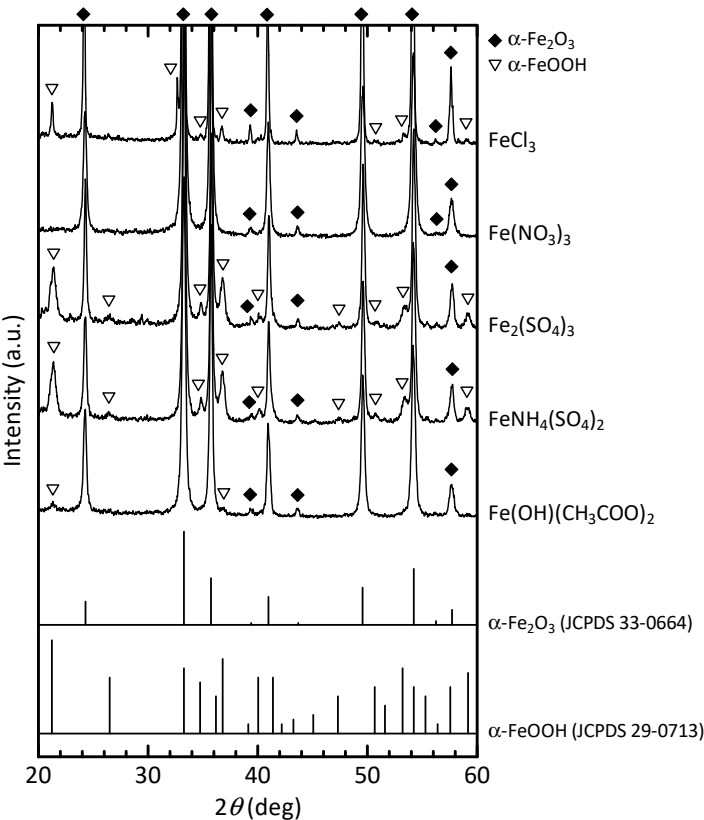

**Figure 2.** XRD patterns of samples prepared using $FeCl_3$, $Fe(NO_3)_3$, $Fe_2(SO_4)_3$, $FeNH_4(SO_4)_2$, and $Fe(OH)(CH_3COO)_2$, as the iron source.

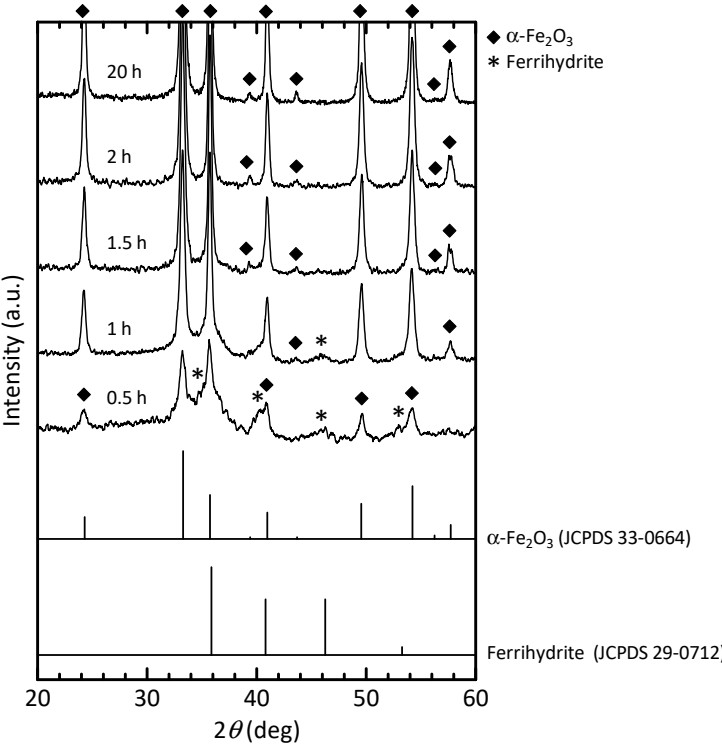

**Figure 3.** Effect of hydrothermal treatment time on phase evolution of samples prepared using $Fe(NO_3)_3$.

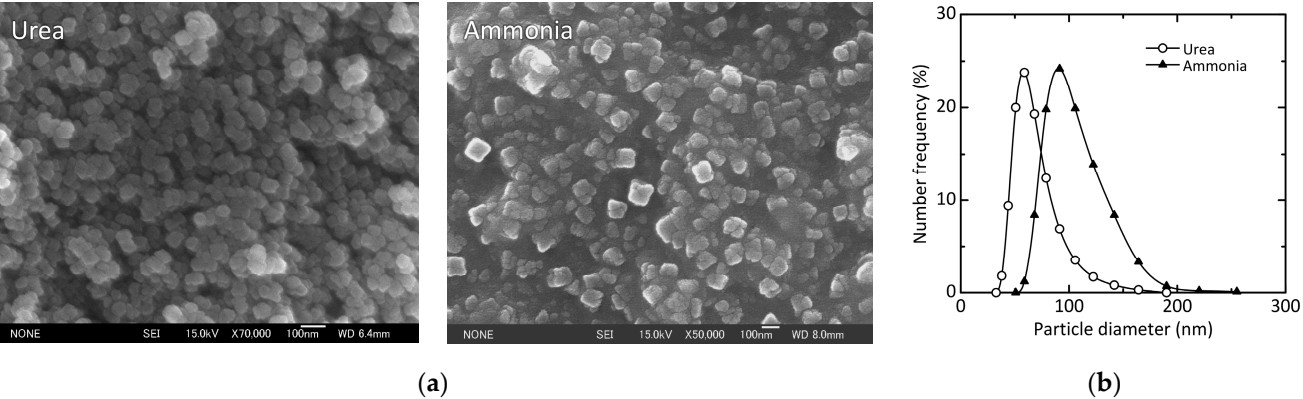

**Figure 4.** (**a**) SEM images and (**b**) particle-size distributions of samples prepared using urea and ammonia as the precipitating agents by hydrothermal treatment for 2 h.

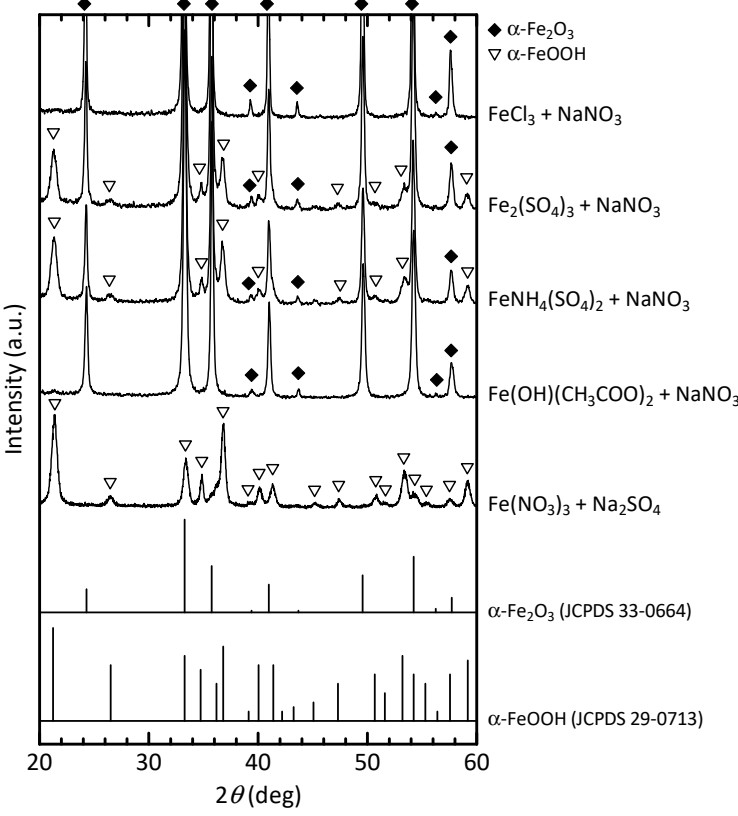

**Figure 5.** Effects of addition of $NO_3^-$ and $SO_4^{2-}$ ions to the starting solutions of different iron salts on the hydrothermal formation of $\alpha$-$Fe_2O_3$.

However, when the $Fe(NO_3)_3$ solution contained 1.5 mmol of $Na_2SO_4$, in which the molar ratio of $SO_4^{2-}/NO_3^-$ in the solution was 0.1, the obtained sample comprised mostly $\alpha$-FeOOH, as shown in Figure 5, indicating that $SO_4^{2-}$ ions may have considerably reduced the reaction rate of $\alpha$-$Fe_2O_3$ formation. In particular, the reactivity and stability of intermediate ferric hydroxide may be varied by coexisting $SO_4^{2-}$ ions [32,33]. Accordingly, the coexisting anions in the solution were found to greatly affect the hydrothermal formation of $\alpha$-$Fe_2O_3$. $NO_3^-$ and $SO_4^{2-}$ ions can especially promote and inhibit the $\alpha$-$Fe_2O_3$ formation reaction in this process, respectively. In addition, the effects of $SO_4^{2-}$ ions may be stronger than those of $NO_3^-$ ions. The results suggested that the stability of iron complexes in the solution plays an important role in the hydrothermal formation of $\alpha$-$Fe_2O_3$. However, the detailed mechanism remains unknown and requires further study.

The crystallite sizes of samples with a single $\alpha$-Fe$_2$O$_3$ phase prepared using Fe(NO$_3$)$_3$, NaNO$_3$-added FeCl$_3$, and NaNO$_3$-added Fe(OH)(CH$_3$COO)$_2$ solutions were calculated using the Scherrer equation. The average values of crystallite size, which were calculated using the diffraction intensity peaks observed at $2\theta \approx 24.1°$, $33.2°$, $35.6°$, $40.9°$, $49.5°$, $54.1°$, $62.4°$, and $64.0°$ corresponding to the (012), (104), (110), (113), (024), (116), (214), and (300) crystal planes of $\alpha$-Fe$_2$O$_3$, respectively, the crystallinity index *CI* (%), which is defined by Equation (8),

$$CI = \frac{I_t - I_a}{I_t} \times 100 \tag{8}$$

where $I_t$ $(-)$ is the total intensity for $\alpha$-Fe$_2$O$_3$ (110) plane and ferrihydrite (110) plane at $2\theta \approx 35.7°$ and $I_a$ $(-)$ is the intensity for ferrihydrite (113) plane at $2\theta \approx 46.3°$ as amorphous phase, and the median diameter of number-basis distribution of particle sizes are summarized in Table 1. The SEM images are also shown in Figure 6. Although the crystallinity index was almost the same (98–99%), the crystallite size and particle diameter varied depending on the coexisting anions in the solution, suggesting that their control is possible by using the coexisting anions as an operating factor.

**Table 1.** Average crystallite size, crystallinity index, and median diameter of $\alpha$-Fe$_2$O$_3$ nanoparticles obtained after hydrothermal treatment for 20 h.

| Starting Material | | Average Crystallite Size (nm) | Crystallinity Index (%) | Median Diameter (nm) |
|---|---|---|---|---|
| **Iron Source** | **Additive** | | | |
| Fe(NO$_3$)$_3$ | – | 24.1 | 98.1 | 53.2 |
| FeCl$_3$ | NaNO$_3$ | 46.8 | 99.0 | 125.4 |
| Fe(OH)(CH$_3$COO)$_2$ | NaNO$_3$ | 30.8 | 98.6 | 55.5 |

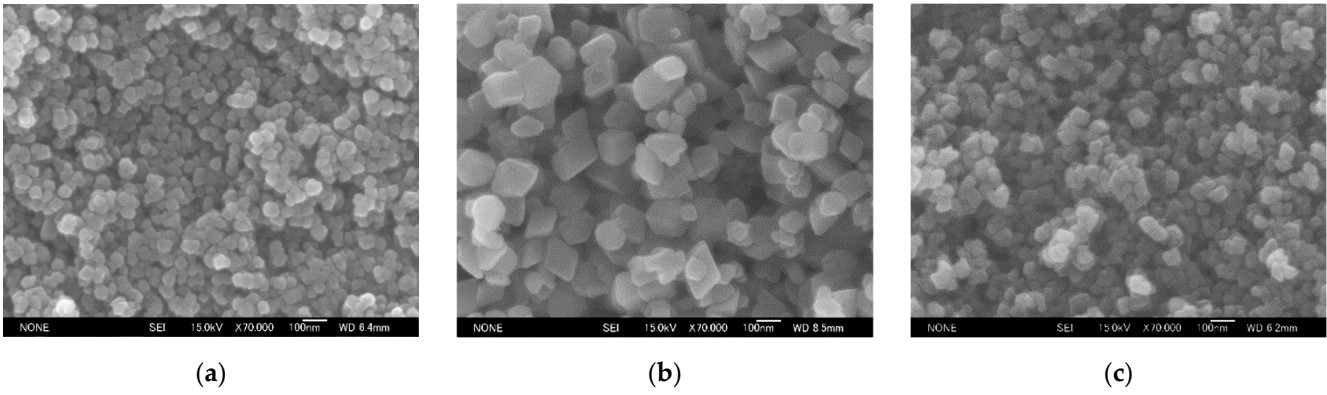

**Figure 6.** SEM images of $\alpha$-Fe$_2$O$_3$ nanoparticles prepared with (**a**) Fe(NO$_3$)$_3$, (**b**) NaNO$_3$-added FeCl$_3$, and (**c**) NaNO$_3$-added Fe(OH)(CH$_3$COO)$_2$ solutions by hydrothermal treatment for 20 h.

*3.2. Adsorption Studies of Congo Red*

As shown in Figure 7a, the removal efficiency increased with increasing adsorbent dosage due to an increase in the number of active adsorption sites. In contrast, with a decrease in pH, the removal efficiency increased (Figure 7b), which was similar to the results reported in the literature [51]. In particular, when pH $\leq 4.6$, the removal efficiency reached above 80% and the adsorbent showed good adsorption performance. This may be attributed to electrostatic interaction between the adsorbent and the anionic dye molecules due to the positive surface potential of adsorbent [51], as seen in the pH dependence of zeta potential (Figure 7b).

The change in the adsorbed amount $q_t$ with contact time is illustrated in Figure 8. The adsorbed amount rapidly increased in the initial stage and approximately reached equilibrium after several hours. To analyze the adsorption kinetics, several representative models, i.e., pseudo-first-order model, pseudo-second-order model, intra-particle diffusion

model, and Elovich model [54], which are expressed by Equations (9), (10), (11), and (12), respectively, were applied to the experimental data shown in Figure 7.

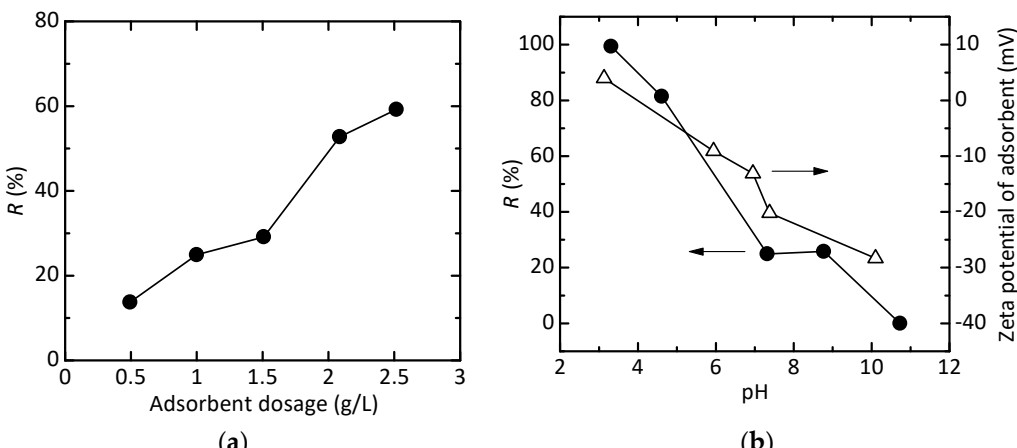

**Figure 7.** Effect of (**a**) adsorbent dosage and (**b**) solution pH on the removal efficiency. The closed circles and open triangles in (**b**) indicate the removal efficiency *R* and the zeta potential, respectively.

$$q_t = q_e \left(1 - e^{-k_1 t}\right) \tag{9}$$

$$q_t = \frac{k_2 q_e^2 t}{1 + k_2 q_e t} \tag{10}$$

$$q_t = k_i t^{0.5} + c \tag{11}$$

$$q_t = \frac{1}{\beta} \ln(\alpha \beta t) \tag{12}$$

$k_1$ (h$^{-1}$), $k_2$ (g/(mg·h)), and $k_i$ (mg/(g·h$^{0.5}$)) are the rate constants of the pseudo-first-order, pseudo-second-order, and intra-particle diffusion models, respectively. $c$ (mg/g) is a constant. $\alpha$ (mg/(g·h)) and $\beta$ (g/mg) are the initial sorption rate and the constant related to the extent of surface coverage and activation energy for chemisorption, respectively. The nonlinear fitting results using Equations (9)–(12) and the determined parameters, which were obtained by the Solver function of Microsoft Excel 2016 (Microsoft Corporation, Redmond, WA, USA), are shown in Figure 8 and Table 2, respectively. The experimental data fit well with the Elovich model, which suggest chemical adsorption process and the activation energy increased with adsorption time [55].

Using the relationship between $q_e$ and $C_e$ shown in Figure 9, the adsorption isotherm was analyzed using the Langmuir model, the Freundlich model, and the Temkin model, which are described by Equations (13), (14), and (15), respectively.

$$q_e = \frac{q_m K_L C_e}{1 + K_L C_e} \tag{13}$$

$$q_e = K_F C_e^{1/n} \tag{14}$$

$$q_e = q_T \ln(A_T C_e) \tag{15}$$

where $q_m$ (mg/g) is the maximum monolayer adsorption capacity and $K_L$ (L/mg) is the Langmuir equilibrium constant. $K_F$ (mg/(g·(mg/L)$^{1/n}$)) and $n$ (−) are the Freundlich constants related to the adsorption capacity and the intensity of adsorption, respectively. $A_T$ (L/mg) and $q_T$ (mg/g) are the adsorption equilibrium constant of solute on solid surface [56] and the surface capacity for contaminant adsorption per unit binding energy [57], respectively. As shown in Figure 9 and Table 3, the experimental data were well-described by the Langmuir model, which suggested the homogeneous monolayer adsorption. Here, the comparison of the maximum monolayer adsorption capacities with

some literature values is summarized in Table 4. Although no additional treatments for enhancing the adsorption performance such as surface modification were performed, our hematite nanoparticles, which were synthesized by the simple process using inexpensive raw materials, have relatively good adsorption properties and can be a promising candidate as an adsorbent for Congo red.

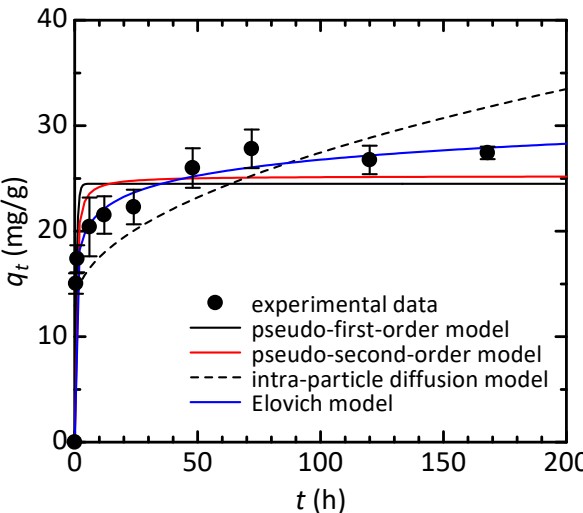

**Figure 8.** Adsorption kinetics plots: pseudo-first-order model, pseudo-second-order model, intra-particle diffusion model, and Elovich model.

**Table 2.** Values of parameters for adsorption kinetic models.

| Kinetic Model | Parameter | Value |
|---|---|---|
| Pseudo-first-order | $q_e$ (mg/g) | 24.5 |
| | $k_1$ (h$^{-1}$) | 1.55 |
| | $R^2$ | 0.902 |
| Pseudo-second-order | $q_e$ (mg/g) | 25.2 |
| | $k_2$ (g/(mg·h)) | 0.0928 |
| | $R^2$ | 0.934 |
| Intra-particle diffusion | $c$ (mg/g) | 12.9 |
| | $k_i$ (mg/(g·h$^{0.5}$)) | 1.45 |
| | $R^2$ | 0.607 |
| Elovich | $\alpha$ (mg/(g·h)) | $4.55 \times 10^3$ |
| | $\beta$ (g/mg) | 0.457 |
| | $R^2$ | 0.959 |

The adsorption process was thermodynamically analyzed using the following equations [52]:

$$\Delta G^\circ = -RT \ln K_e^\circ \tag{16}$$

$$\ln K_e^\circ = \frac{\Delta S^\circ}{R} - \frac{\Delta H^\circ}{RT} \tag{17}$$

where $\Delta G^\circ$ (J/mol), $\Delta S^\circ$ (J/(mol·K)), and $\Delta H^\circ$ (J/mol) are the Gibbs free energy, the standard entropy, and the standard enthalpy, respectively. $K_e^\circ$ (−) is the thermodynamic equilibrium constant and is obtained by Equation (18) [52].

$$K_e^\circ = K_L M c_{ad} \tag{18}$$

where $K_L$ (L/mg) is the Langmuir equilibrium constant of adsorption, which was determined from the isotherm data obtained at different temperatures; $M$ (mg/mol) is the molecular weight of adsorbate; and $c_{ad}$ (mol/L) is the standard concentration of adsorbate,

which is defined as 1 mol/L. The results of linear fitting with Equation (17) are shown in Figure 10 and Table 5. Although it is difficult to evaluate the adsorption performance of an adsorbent based on the magnitude of the values of $\Delta G°$, $\Delta S°$, and $\Delta H°$, it is noteworthy that their values were negative, which indicates that the adsorption process is feasible and spontaneous, decreases in randomness at the adsorbent/dye interface, and is exothermic. In general, when such requirements are satisfied, the adsorption can be favorable. Therefore, using our $\alpha$-$Fe_2O_3$ nanoparticles as the adsorbent for removal of Congo red is practically applicable.

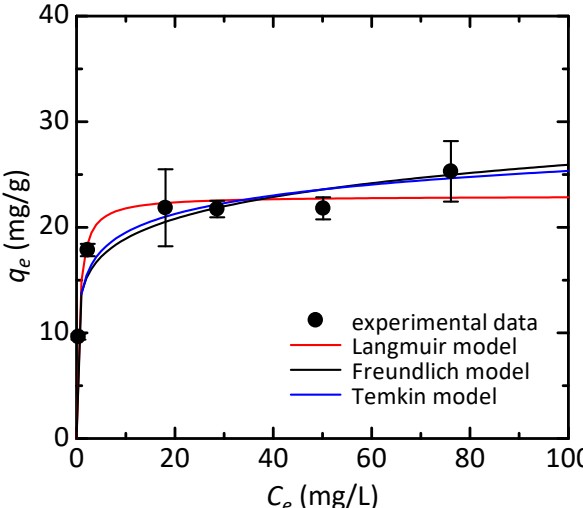

**Figure 9.** Adsorption isotherm of Congo red dye onto the $\alpha$-$Fe_2O_3$ nanoparticles using the Langmuir model, the Freundlich model, and the Temkin model.

**Table 3.** Values of parameters for adsorption isotherm models.

| Isotherm Model | Parameter | Value |
|---|---|---|
| Langmuir | $q_m$ (mg/g) | 23.0 |
| | $K_L$ (L/mg) | 1.86 |
| | $R^2$ | 0.944 |
| Freundlich | $K_F$ (mg/(g·(mg/L)$^{1/n}$)) | 13.8 |
| | $n$ (−) | 7.32 |
| | $R^2$ | 0.885 |
| Temkin | $A_T$ (L/mg) | 221 |
| | $q_T$ (mg/g) | 2.53 |
| | $R^2$ | 0.926 |

**Table 4.** Comparison of the maximum adsorption capacities calculated from Langmuir model.

| Adsorbent | $q_m$ (mg/g) | Reference |
|---|---|---|
| Porous $\alpha$-$Fe_2O_3$ nanorod | 57.2 | [51] |
| $Fe_3O_4$@$SiO_2$@Zn–TDPAT | 17.7 | [58] |
| MgFeAl LDHs | 14.8 | [59] |
| NiFeTi LDHs | 30.0 | [60] |
| $MnFe_2O_4$ | 25.8 | [61] |
| Fe–Zn bimetallic nanoparticles | 28.6 | [62] |
| $\alpha$-$Fe_2O_3$ nanoparticles | 23.0 | This work |

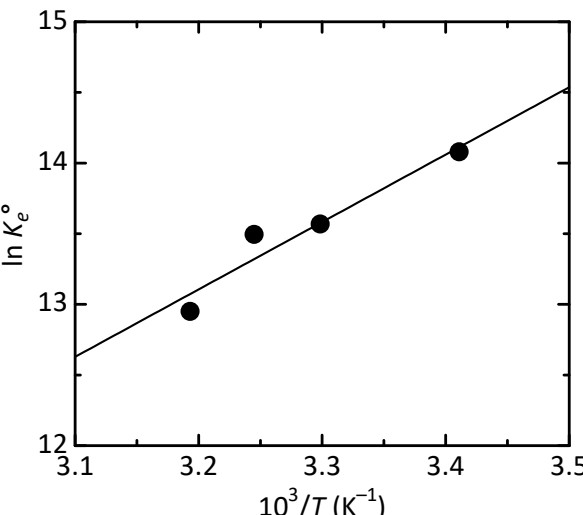

**Figure 10.** Plot of ln $K_e°$ against $1/T$ for estimation of thermodynamics parameters ($R^2 = 0.928$).

**Table 5.** Thermodynamic parameters for Congo red adsorption on the $\alpha$-Fe$_2$O$_3$ nanoparticles.

| Temperature (K) | $\Delta G°$ (kJ/mol) | $\Delta S°$ (J/(mol·K)) | $\Delta H°$ (kJ/mol) |
|---|---|---|---|
| 293 | −34.40 | | |
| 303 | −34.22 | | |
| 308 | −34.13 | −18.0 | −39.66 |
| 313 | −34.04 | | |

For investigating the reusability of the adsorbent, the regeneration experiment was performed. After adsorption for more than 12 h under the typical conditions except for the initial concentration (10 mg/L in this investigation), the spent adsorbent was isolated from the dye solution and heated at 673 K in air for 2 h [63] as an example. The adsorption–regeneration was repeated three times under the same conditions. As shown in Figure 11, relatively large adsorption capacities were observed for the spent adsorbent even after the regeneration, although it had a tendency to decrease with increasing the cycle number. The result suggests that the adsorbent is reusable after the conditions for regeneration are optimized for maintaining the adsorption performance in the recycling.

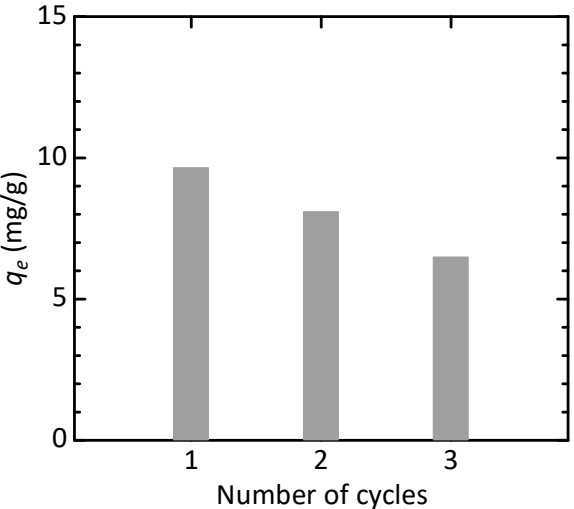

**Figure 11.** Equilibrium adsorption capacity of regenerated $\alpha$-Fe$_2$O$_3$ nanoparticles.

## 4. Conclusions

In the hydrothermal synthesis of crystalline $\alpha$-Fe$_2$O$_3$ nanoparticles via urea hydrolysis, the anions contained in the starting solution greatly affected the rate of $\alpha$-Fe$_2$O$_3$ formation. In particular, NO$_3^-$ ion promoted the formation reaction, which may be due to the formation of a 6-line ferrihydrite intermediate. This can contribute to the rapid synthesis of $\alpha$-Fe$_2$O$_3$ nanoparticles. In contrast, when using FeCl$_3$, Fe$_2$(SO$_4$)$_3$, FeNH$_4$(SO$_4$)$_2$, or Fe(OH)(CH$_3$COO)$_2$ as the iron source, the products contained the intermediate $\alpha$-FeOOH in addition to $\alpha$-Fe$_2$O$_3$. The addition of NaNO$_3$ to the FeCl$_3$ and Fe(OH)(CH$_3$COO)$_2$ solutions provided a single $\alpha$-Fe$_2$O$_3$ phase. However, the addition of NaNO$_3$ to the Fe$_2$(SO$_4$)$_3$ and FeNH$_4$(SO$_4$)$_2$ solutions resulted in no changes in phase evolution. Accordingly, SO$_4^{2-}$ ions tended to inhibit the reaction, which may be due to the formation of a relatively stable iron complex. Furthermore, the effects of SO$_4^{2-}$ ions may be stronger than those of NO$_3^-$ ions. The average crystallite size and median diameter of $\alpha$-Fe$_2$O$_3$ nanoparticles prepared using the Fe(NO$_3$)$_3$, NaNO$_3$-added FeCl$_3$, and NaNO$_3$-added Fe(OH)(CH$_3$COO)$_2$ solutions also depended on the coexisting anions. Therefore, selecting the coexisting anions appropriately can contribute to controlling the growth of $\alpha$-Fe$_2$O$_3$ nanoparticles.

The $\alpha$-Fe$_2$O$_3$ nanoparticles prepared with Fe(NO$_3$)$_3$ were used as the adsorbent for removal of Congo red dye. The adsorption kinetics and isotherm followed the Elovich and Langmuir models, respectively. The adsorption thermodynamic study revealed that the adsorption process was feasible and practically applicable. The results demonstrated that the $\alpha$-Fe$_2$O$_3$ nanoparticles synthesized by the simple process using the inexpensive raw materials have relatively good adsorption properties without additional functionalization, such as surface modification, and are a promising candidate as adsorbents for Congo red.

**Author Contributions:** Conceptualization, T.I.; methodology, T.I.; validation, T.O. and T.I.; formal analysis, T.O. and T.I.; investigation, T.O., M.F. and T.I.; data curation, T.O., M.F. and T.I.; writing—original draft preparation, T.O. and T.I.; writing—review and editing, T.I.; visualization, T.O. and T.I.; supervision, T.I.; project administration, T.I. All authors have read and agreed to the published version of the manuscript.

**Funding:** This research received no external funding.

**Institutional Review Board Statement:** Not applicable.

**Informed Consent Statement:** Not applicable.

**Data Availability Statement:** Not applicable.

**Conflicts of Interest:** The authors declare no conflict of interest.

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
