# Peer review of "Effects of Coexisting Anions on the Formation of Hematite Nanoparticles in a Hydrothermal Process with Urea Hydrolysis and the Congo Red Dye Adsorption Properties"

_2674-0516, doi:10.3390/powders2020020_

Round 1
Reviewer 1 Report
The paper is well-written and presents a complete and holistic study of the synthesis of adsorbent nanoparticles and their application for dye removal. The manuscript can be accepted after minor revision. I have included some comments for the authors to consider.
If possible, the authors could expand on the reactor used. Was it a high-pressure digestor (possibly from Berghof?). Including the total volume of the autoclave, how much of the volume was occupied by reagents, and the expected pressure within the reactor would be beneficial for the reader.
Was there a specific reason for the selection of Congo Red? If it is due to its carcinogenic nature, it should be mentioned.
Revise the use of “proper amount of time”.
If possible, can the observed XRD patterns be compared to what is available in the literature?
A legend should be added to Figures 1 and 2 to show which symbol is representative of which phase.
Revise the way how synthesis routes and the obtained particles are presented in Fig. 3. E.g. “… prepared using (a, c) urea…”; the reason being is that (a) here refers to Fig. 3a, whereas (c) refers to a curve and a figure labeled (c,d). It is advised to refer to the particle size distribution curves as Figure 3c, and allocated dissimilar labeling to the curves (e.g., Curve 1 = urea, Curve 2 = ammonia).
Similarly in Fig. 4; referring to Fig. 4e as a curve pattern should be avoided. Each curve has to be labeled independently for its figure number. Moreso, Fig. 4 should have a legend and each of the symbols allocated to a certain phase.
The authors provide a good explanation of how the results shown in Figure 9 and Table 5 are calculated. It is advised that the effect of each of the observations is discussed with some detail, e.g., is spontaneous adsorption preferred? Can the level of spontaneity be quantified? What is the importance of a decrease in randomness at the adsorbent/dye interface? Does the exothermic observation result in possible complications the prepared adsorbent is used in the “real world”?
Reviewer 2 Report
Reconsider after major revision.
(1) In the introduction part, the authors spend a lot of space describing the α-Fe2O3 nanoparticle. However, more descriptions of Congo red should be added and related references about adsorption also need cite.
(2) The error bar of all the data needs to add.
(3) Elovich adsorption kinetics models are recommended to add. Please refer to the paper (10.1016/j.cej.2022.138127).
(4) In adsorption isotherm, more data should be added. Besides, Temkin models should be added. Please refer to the paper (10.1016/j.cej.2022.138127).
(5) The reusability of materials is also very important, please add regeneration experiments.
(6) In the effects of pH solution on Congo red adsorption, the zeta potential should add, which helps explore the adsorption mechanism.
(7) In table 4, compared with other adsorption materials, the adsorption ability of α-Fe2O3 nanoparticles in this work is common. What is the significance of this work?
(8) The wording of CONCLUSION is very similar to that of ABSTRACT. Please revise.
Reviewer 3 Report
In this paper synthesized α-Fe2O3 nanoparticles and the adsorption properties towards Congo red dye was investigated. The effects of coexisting anions on the formation of α-Fe2O3 nanoparticles by using a hydrothermal process were investigated in the first time.
1. In the Introduction section should be explained in details, why adsorption properties of hematite nanoparticles were investigated namely towards to Congo red dye.
2. In subsection 2.2 should be presented chemical structure of Congo red dye.
3. Results and discussion
- According to XRD results must be calculated not only crystallite size and Segal crystallinity index.
- The authors wrote the following:
The non-linear fitting results using Equations (8)–(10) and the determined parameters are shown in Figure 7 and Table 2, respectively the (Page 7, Lines 206-208). By using which software, the results were found?
- The same comments according to the Figure 8.
Round 2
Reviewer 1 Report
The manuscript has gained clarity and scientific soundness and can now by accepted for publication.
Reviewer 2 Report
ACCEPT
Reviewer 3 Report
can be accepted